# TTSS-2 virulence drives inflammatory destruction of the gut epithelial barrier and modulates inflammatory response profiles in the *Salmonella*-infected mouse gut

Ursina Enz[1], Ersin Gül[1,¤a], Luca Maurer[1], Kamilė Čerepenkaitė[1], Jemina Huuskonen[1], Stefan A. Fattinger[1,2,¤b], Wolf-Dietrich Hardt [1]*

**1** Department of Biology, Institute of Microbiology, ETH Zürich, Zürich, Switzerland, **2** Science for Life Laboratory, Department of Medical Biochemistry and Microbiology, Uppsala University, Uppsala, Sweden

☺ These authors contributed equally to this work.
¤a Current address: Department of Immunology, University of Washington School of Medicine, Seattle, Washington, USA
¤b Current address: Division of Immunology and Molecular Medicine, Department of Molecular and Cell Biology, University of California, Berkeley, California, USA.
* hardt@micro.biol.ethz.ch

## Abstract

*Salmonella enterica* serovar Typhimurium (*S.* Tm) employs type III secretion system 1 and 2 (TTSS-1 and TTSS-2) to infect host tissues. In orogastric infections, both TTSSs manipulate host responses, increasing mucosal pathogen loads and eliciting inflammation. However, we still do not fully understand how virulence and inflammatory enteropathy are interconnected. Here, we investigate whether TTSS-2–dependent virulence contributes to epithelial barrier disruption and delineate its role in shaping inflammatory response profiles in the mouse gut. Using wild-type and TTSS-2 mutant *S.* Tm strains in antibiotic-pretreated mouse models, we demonstrate that intestinal epithelial destruction is promoted by TTSS-2 virulence. This effect is observed in both wild-type and immune-deficient C57BL/6J mice. Transcriptomic profiling together with immunofluorescence microscopy analysis reveals that wild-type *S.* Tm triggers a distinct, yet amplified immune response compared to a TTSS-2 mutant which is characterized by enhanced phagocyte recruitment and a unique transcriptional signature. These findings underscore the role of TTSS-2–mediated virulence in *S.* Tm gut infection, shaping distinct inflammatory microenvironments with potential implications for host-pathogen interaction studies.

## Author summary

The intestinal lining forms a crucial barrier that protects the body from potentially harmful microbes in the gut. Enteric pathogens like *Salmonella enterica* serovar

**Data availability statement:** The sequencing data for this study have been deposited in the European Nucleotide Archive (ENA) under accession number PRJEB91165 (https://www.ebi.ac.uk/ena/browser/view/PRJEB91165). Additional analyses of the data are deposited in Zenodo (https://doi.org/10.5281/zenodo.15783423). All other relevant data are within the paper and its Supporting information files.

**Funding:** This work has been funded by grants from the Swiss National Science Foundation (310030_192567 and 10.001.588) to WDH. EG received a PhD stipend from the Monique Dornonville de la Cour Foundation, granted to WDH. The funders had no role in study design, data collection and analysis, decision to publish, or preparation of the manuscript.

**Competing interests:** The authors have declared that no competing interests exist.

Typhimurium (*S*. Tm) – a major cause of foodborne illness – can overcome this barrier to invade host tissues and trigger intestinal inflammation by specialized virulence strategies. A key player in this process is the type III secretion system 2 (TTSS-2), which enables the pathogen to manipulate host cells during infection. However, how this virulence factor contributes to gut tissue damage and modulates the immune response remains unclear. In this study, we used antibiotic-pretreated mouse models to investigate the role of TTSS-2 in driving intestinal pathology. We found that TTSS-2 activity promotes destruction of the gut epithelium and induces a stronger, qualitatively different immune response compared to a TTSS-2–deficient strain. This response includes increased recruitment of immune cells and a distinct transcriptional profile in the gut tissue. Our results suggest that TTSS-2–mediated virulence not only enhances bacterial survival and replication but also actively shapes the inflammatory landscape of the gut. These findings improve our understanding of how specific bacterial virulence factors contribute to inflammation and epithelial barrier disruption during enteric infection.

## Introduction

*Salmonella enterica* serovar Typhimurium (*S*. Tm) is a widely studied model enteric pathogen for invasive bacterial gut infections [1]. It is used to investigate pathogenic mechanisms such as bacterial invasion, immune evasion, and intracellular survival. Importantly, well-established mouse infection models allow for controlled analysis of gut inflammation, providing a versatile platform to explore both host-pathogen and microbe-microbe interactions under inflammatory conditions in the intestinal environment [1–13].

Under homeostatic conditions, the complex gut microbiota confer colonization resistance, protecting against enteric pathogens like *S*. Tm [3,14]. This poses a significant technical challenge for virulence studies. Therefore, murine models with perturbed microbiota, such as streptomycin-pretreated mice, are widely used to study *S*. Tm pathogenesis [1]. In these models, using its type-three secretion systems-1 and -2 (TTSS-1 and -2), *S*. Tm robustly induces disease by efficiently invading intestinal epithelial cells (IECs), followed by transmigrating into the underlying lamina propria, and spreading to systemic organs [15–17].

Invasion of the epithelium is promoted by the *Salmonella* pathogenicity island-1 (SPI-1), a genomic region encoding for the TTSS-1 and various effector proteins [18–20]. SPI-1-dependent insult on epithelial cells leads to extrusion of infected IECs. In mice, this controlled epithelial-intrinsic response is dependent on the NAIP/NLRC4 inflammasome [21,22]. Continued, pronounced SPI-1-dependent assault on the epithelium appears to drive less regulated IEC extrusion in later stages of infection, which is thought to be NLRC4-independent and further exacerbated by TNF [23]. In response to IEC extrusion, crypt stem cells ramp up proliferation, leading to crypt hyperplasia [1,24,25].

After invasion of gut epithelial cells and colonization of the gut lumen, *S*. Tm disseminates deeper into the gut tissue and to systemic sites ($\approx 10^4$-$10^5$ CFU in the spleen by day 3–4 post infection (p.i.) [15]). Once intracellularly, *S*. Tm expresses SPI-2 genes (encoding TTSS-2 and various effector proteins) that are important for intracellular survival and replication of *S*. Tm [15,16,26–35]. At this stage, signs of full-blown inflammation are observable in the cecum tissue. Histopathology of inflamed cecum tissue is characterized by submucosal edema and reduced number of goblet cells [1]. Also, neutrophils are recruited to the lamina propria and can be found in the gut lumen as aggregates [21,36–38].

*S*. Tm without functional TTSS-2 demonstrates reduced efficiency in colonizing gut tissue and systemic organs compared to wild-type strains [15,30]. However, infections with TTSS-2 mutant *S*. Tm still exhibit multiple signs of gut tissue inflammation [39]. Although several studies that use TTSS-2 mutant *S*. Tm for infection experiments show that TTSS-2 virulence adds to gut disease [15,38–41], we still lack a complete picture of the impact of TTSS-2 virulence to gut inflammation, in particular at days 2–4 of *Salmonella* gut infection.

Here, we performed a systematic comparison of wildtype *S*. Tm and TTSS-2 mutant *S*. Tm to characterize the contribution of *S*. Tm's TTSS-2 virulence to gut inflammation. Our data confirm previous findings that *S*. Tm's TTSS-2 virulence exacerbates gut inflammation in the late stages of infection in streptomycin-pretreated mice [13,15,38–41]. Extending these findings, we show that exacerbated gut inflammation is not only characterized by epithelial barrier destruction – driven by increased epithelial cell expulsion, reduced epithelial cell proliferation, and heightened neutrophil and macrophage recruitment – but also by significant transcriptomic changes in the cecum tissue that reflect these phenotypic observations.

Notably, it is the direct comparison between wild-type *S*. Tm and the TTSS-2 mutant in wild-type mice that allows us to characterize the distinct inflammatory responses, providing deeper insights into how and to what extent TTSS-2 contributes specifically to gut pathology. In parallel, analysing wild-type *S*. Tm and the TTSS-2 mutant in immunocompromised mice further underscores the interrelationship between pathogen virulence factors and host immune competence, thereby highlighting the pivotal role of TTSS-2 virulence in driving *S*. Tm-mediated gut inflammation.

## Results

### The collapse of the epithelial barrier in NLRC4-deficient mice is TTSS-2-dependent

In streptomycin-pretreated mice, the NAIP/NLRC4 inflammasome limits local and systemic *S*. Tm loads in the gut tissue as a first line defence by promoting expulsion of infected enterocytes [21,23,42]. In the absence of this defence, i.e., in NLRC4-deficient mice, the tissue loads are exacerbated, leading to the loss of gut epithelial integrity within 72 h following infection with wild-type *S*. Tm (*S*. Tm^WT) [23]. However, it remained unclear whether such disease pathology also occurs during infections with *S*. Tm mutants lacking a functional TTSS-2. To address this, we investigated the contribution of TTSS-2 virulence to disease pathology using a *S*. Tm strain lacking *ssaV* (Δ*ssaV*, denoted as *S*. Tm^SPI2 [43]), a critical gene of the TTSS-2 apparatus encoded on SPI-2.

We compared *S*. Tm^WT or *S*. Tm^SPI2 infection (for 72 h) in streptomycin-pretreated NLRC4-deficient mice and their heterozygous littermates to observe the development of gut disease and changes in epithelial barrier integrity (Fig 1A). Infection of streptomycin-pretreated NLRC4-deficient mice and heterozygous controls with *S*. Tm^WT confirmed prior observations [23]: while luminal colonization was comparable across genotypes (S1A Fig), NLRC4-deficient mice exhibited elevated *S*. Tm^WT colony forming units (CFUs) in cecum tissue and mesenteric lymph nodes (mLN) compared to heterozygous controls (S1C and S1D Fig). As previously shown by Fattinger et al. [23], fluorescence microscopy confirmed severe epithelial disruption and reduced epithelial regeneration capacity in NLRC4-deficient mice by 72 h p.i., as indicated by lower numbers of Ki67-positive cells (Fig 1B and 1C) and slightly elevated histopathology scores (S1G and S1H Fig). To assess the role of TTSS-2 in driving this pathology, we infected NLRC4-deficient and heterozygous control mice with *S*. Tm^SPI2 (Fig 1A). While luminal *S*. Tm^SPI2 colonization remained similar between NLRC4-deficient and control mice, cecum

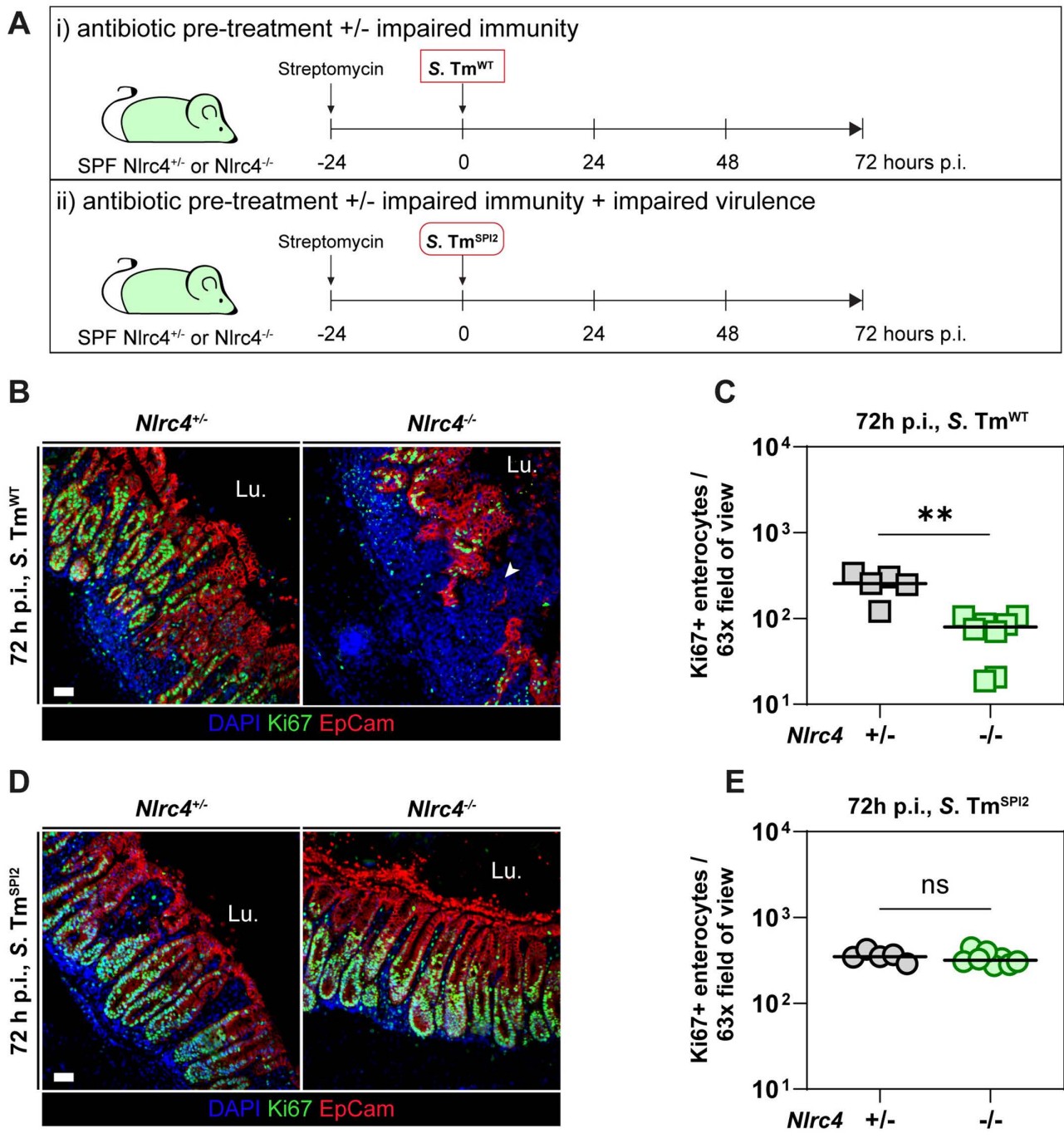

**Fig 1. In streptomycin-pretreated NLRC4-deficient mice, epithelial barrier disruption upon *S.* Tm infection is driven by TTSS-2-dependent virulence.** (a–e) N*lrc4*+/- and *Nlrc4*-/- mice were streptomycin-pretreated, infected with *S.* Tm^WT (SB300) or *S.* Tm^SPI2 (Δ*ssaV*), and sacrificed at 72 h p.i. (a) Experimental setup. (b) Representative micrographs of *S.* Tm^WT-infected *Nlrc4*+/- and *Nlrc4*-/- cecum tissue sections, stained for cell proliferation marker Ki-67 at 72 h p.i. Arrowhead indicates gap in epithelium. Lu.: Lumen. Scale bar: 50 µm. (c) Microscopy-based quantification of Ki-67 positive epithelial cells per 63x field of view in (b). (d) Representative micrographs of *S.* Tm^SPI2-infected *Nlrc4*+/- and *Nlrc4*-/- cecum tissue sections, stained for cell proliferation marker Ki-67 at 72 h p.i. Lu.: Lumen. Scale bar: 50 µm. (e) Microscopy-based quantification of Ki-67 positive epithelial cells per 63x field of view in (d). (c, e) Each point represents average of one mouse. Line at median. Mann-Whitney *U* test (* $p < 0.05$, ** $p < 0.01$, ns – not significant).

tissue CFUs tended to be slightly higher and mLN CFUs were 5–10 times higher in NLRC4-deficient animals (S1B–D Fig). However, it needs to be noted that the cecum tissue densities remained approximately 5-fold lower in NLRC4-deficient mice infected with S. Tm^SPI2 compared to the NLRC4-deficient mice infected with S. Tm^WT (S1C Fig), consistent with the well-established role of SPI-2 virulence in promoting intracellular replication of S. Tm [15,26–29,31–35]. These reduced tissue loads may explain why NLRC4-deficient mice infected with S. Tm^SPI2 retained an intact epithelial barrier, showed no defects in regeneration capacity (Fig 1D and 1E) and overall lower histopathology scores than in S. Tm^WT infection. We hypothesize that S. Tm^SPI2 could not surpass the threshold-density in the gut tissue required to trigger TNF-dependent damage exacerbation.

In summary, the epithelial barrier collapse in NLRC4-deficient mice depends on TTSS-2-mediated virulence, as S. Tm^SPI2 fails to elicit comparable pathology in these animals at the investigated time point.

## Epithelial barrier disruption requires TTSS-2 and is not restricted to NLRC4-deficient mice

We observed that gut epithelial disruption following S. Tm infection in NLRC4-deficient mice depends on the presence of the SPI-2 virulence system. This finding led us to question whether this phenomenon is specific to NLRC4-deficiency or if it also occurs in the context of other immune deficiencies with impaired control of bacterial burden during S. Tm infection. One such defence mechanism is mediated by NADPH oxidases, which play an essential role in controlling infections by bacterial pathogens such as *Mycobacterium tuberculosis*, *Aspergillus* spp., and S. Tm by producing reactive oxygen species that have pronounced antimicrobial activity. Notably, TTSS-2 virulence was shown to interfere with NADPH oxidase related defences, as mutants with defective SPI-2 genes featured attenuated virulence in wild-type mice but retained virulence in NADPH oxidase-deficient animals [44]. In this context, CYBB-deficient mice, which lack functional NADPH oxidase activity due to the absence of the Cytochrome b-245 heavy chain encoded by *Cybb*, provide a relevant model to further explore this interaction.

To determine whether S. Tm^SPI2 could: i) proliferate to higher levels in CYBB-deficient mice due to the absence of NADPH oxidase-related defences in phagocytes, and ii) therefore cause epithelial damage in CYBB-deficient mice even without TTSS-2 virulence, we infected CYBB-deficient mice and CYBB-proficient controls with S. Tm^WT or S. Tm^SPI2 (Fig 2A) and analysed cecum tissue integrity by fluorescence microscopy at 72 h p.i.

Indeed, CYBB-deficient mice exhibited higher bacterial loads in systemic organs compared to their CYBB-proficient littermates following infection with either S. Tm^WT or S. Tm^SPI2, while fecal burdens remained similarly high across all groups (S2A–E Fig). However, cecum tissue loads were 10-fold higher in S. Tm^WT than in S. Tm^SPI2 infected animals but did not significantly differ between CYBB-proficient or -deficient mice (S2C Fig). After S. Tm^WT infection, CYBB-deficient mice exhibited impaired epithelial regeneration capacity (Fig 2B and 2C), whereas the CYBB-proficient controls exhibited no defect in epithelial regeneration capacity (judged by Ki67 + cells, Fig 2C). Please note that the number of S. Tm^WT-infected mice in this group was limited, precluding definitive conclusions. Since we observed significant burden in the knockout mice at the end of the infection, we chose not to include further mice into this experiment. Regardless, the observed trend is consistent with the findings in *Nlrc4*^+/- and *Nlrc4*^-/- mice (Fig 1B and 1C). These results extend previous work by showing that multiple innate defence mechanisms help to control tissue-disruptive pathogen expansion in the gut mucosa.

Strikingly, although S. Tm^SPI2 reached bacterial loads 10-fold higher in mLNs and $10^3$-$10^4$-fold higher in the spleens of CYBB-deficient mice compared to their CYBB-proficient littermates (S2D and S2E Fig), the epithelial barrier regeneration capacity remained intact (Fig 2D and 2E). This suggests that, even under these conditions, the mucosal epithelium may remain structurally intact, because S. Tm^SPI2 could not exceed the threshold tissue density required for triggering tissue-disruptive, TNF-dependent immune responses as S. Tm^SPI2 has impaired intracellular replication capacity [15,26–29,31–35]. Indeed, if we plot Ki67 + cell numbers relative to the S. Tm CFUs in the cecum tissue, we can see that higher cecum tissue S. Tm densities correlate with lower Ki67 + enterocyte counts (S2F Fig).

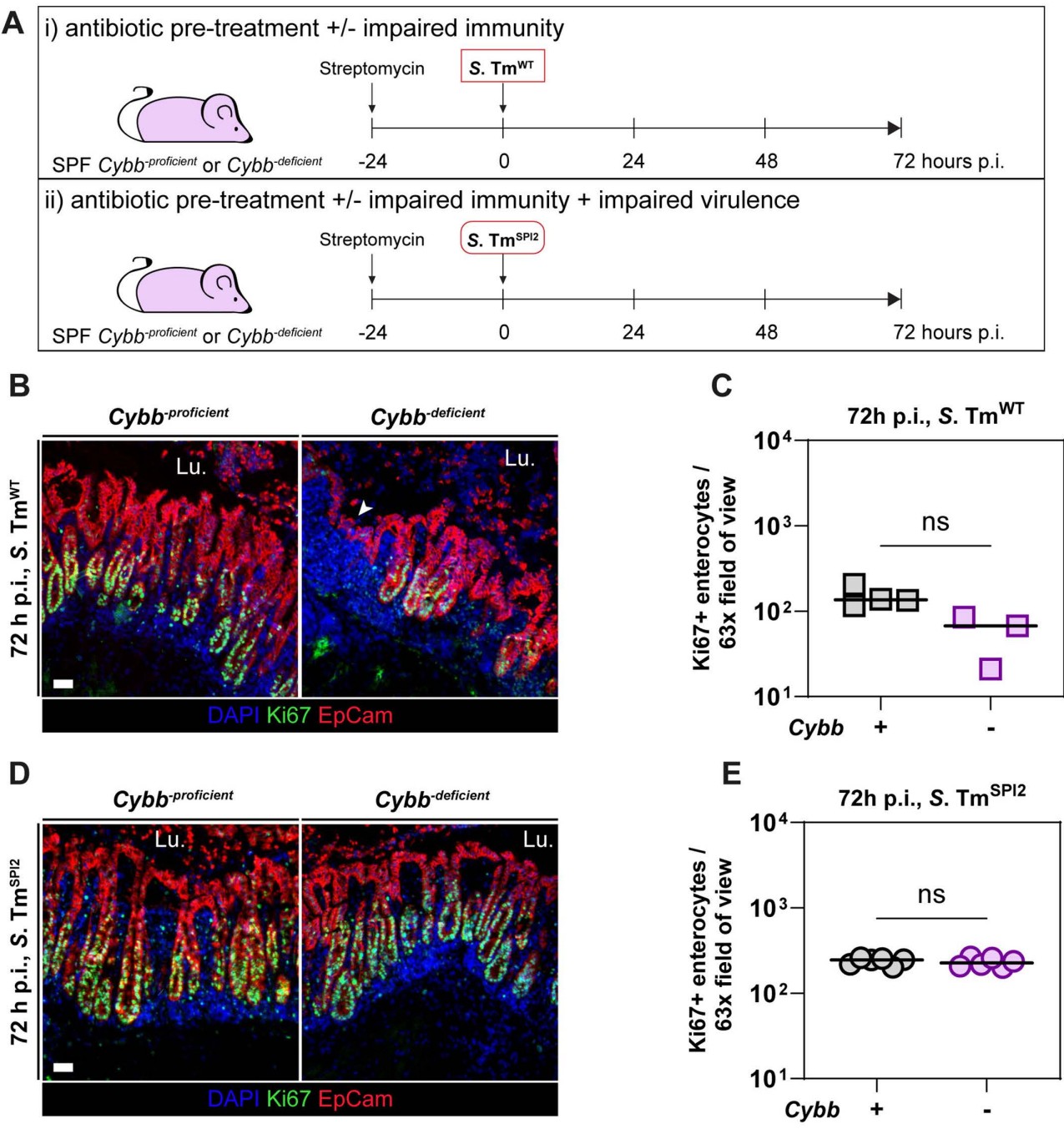

**Fig 2. Epithelial damage in CYBB-deficient mice by 72 h of *S.* Tm infection is TTSS2-dependent.** (a–e) *Cybb*-proficient (*Cybb*-/+; *Cybb*+/+) and *Cybb*-deficient (*Cybb*-/0; *Cybb*-/-) mice were streptomycin-pretreated, infected with *S.* Tm^WT (SB300) or *S.* Tm^SPI2 (Δ*ssaV*) and sacrificed at 72 h p.i. (a) Experimental setup. (b) Representative micrographs of *S.* Tm^WT-infected *Cybb*-proficient and *Cybb*-deficient cecum tissue sections, stained for cell proliferation marker Ki-67 at 72 h p.i. Arrowhead indicates disrupted crypt-villi structure. Lu.: Lumen. Scale bar: 50 μm. (c) Microscopy-based quantification of Ki-67 positive epithelial cells per 63x field of view in (b). (d) Representative micrographs of *S.* Tm^SPI2-infected *Cybb*-proficient and *Cybb*-deficient cecum tissue sections, stained for cell proliferation marker Ki-67 at 72 h p.i. Lu.: Lumen. Scale bar: 50 μm. (e) Microscopy-based quantification of Ki-67 positive epithelial cells per 63x field of view in (d). (c, e) Each point represents average of one mouse. Line at median. Mann-Whitney *U* test (* $p < 0.05$, ** $p < 0.01$, ns – not significant).

Taken together, these findings show that in the absence of NADPH oxidase-related defences, *S*. Tm^SPI2 can proliferate to higher levels at systemic sites and suggest that in mice with impaired immunity (NLRC4 or NADPH oxidase), TTSS-2 virulence is a key driver of severe epithelial pathology observed at 72 h p.i.

**In wild-type C57BL/6J mice, TTSS-2–dependent inflammation disrupts gut epithelial barrier integrity, but this disruption is delayed compared to immune-deficient mice**

In our earlier experiments with immune-deficient mice (NLRC4- and CYBB-deficient, see Figs 1 and 2), we found that TTSS-2 virulence accelerated mucosal histopathology during the first 72h of infection. These findings suggest that in wild-type mice, TTSS-2 triggers an immune response that leads to tissue-disruptive inflammation once the *S*. Tm burden in the gut tissue exceeds a critical threshold. This hypothesis was in line with previous observations: Streptomycin-pretreated C57BL/6J and 129 mice (SPF) infected with *S*. Tm^WT displayed a higher degree of colitis by 96 h p.i. than *S*. Tm^SPI2-infected counterparts [39]. This is also in line with previous studies that compared *S*. Tm^WT and *S*. Tm^SPI2 infection [15,38–41]. However, the mechanisms driving this exacerbation of enteropathy remain unclear. To understand how TTSS-2 virulence contributes to elevated gut inflammation in wild-type mice, we aimed to characterize cecal tissue inflammation at 96 h p.i. with a focus on how TTSS-2-mediated immune responses compromise epithelial barrier integrity and regeneration.

Hence, we infected a new cohort of streptomycin-pretreated C57BL/6J mice with *S*. Tm^WT or *S*. Tm^SPI2 for 96 h (Fig 3A) and in addition, performed microscopy analysis of 96 h-infected mice which we already published in an earlier study [39]. Fecal shedding data indicated that both *S*. Tm^WT or *S*. Tm^SPI2 colonized the gut lumen equally well (S3A Fig). However, cecum tissue *S*. Tm loads at 96 h p.i. were significantly higher in *S*. Tm^WT-infected mice than in the *S*. Tm^SPI2 infected mice (S3B Fig).

To investigate to what degree TTSS-2 virulence impacts gut inflammation over time, we compared the state of the cecum tissue inflammation at 24 h and 96 h after *S*. Tm^WT and *S*. Tm^SPI2 infection. Based on Lipocalin-2 (Lcn-2) levels, both *S*. Tm^WT- and *S*. Tm^SPI2-infection triggered equivalent levels of inflammation at 24 h. In the case of *S*. Tm^WT, the fecal Lcn-2 levels were rising to ≈$10^4$ ng/g feces by 96 h while Lcn-2 levels remained unaltered in *S*. Tm^SPI2-infected mice (≈$10^3$ ng/g feces; Fig 3B). It needs to be noted that Lcn-2 levels above $10^3$ ng/g feces indicate severe gut inflammation [39]. In extension to previous work, we found more dislodged enterocytes in the cecum of *S*. Tm^WT infected mice compared to *S*. Tm^SPI2 infected mice at 96 h p.i. (Fig 3C and 3D). Massive dislodgement of IECs together with the loss of replication capacity (as determined by Ki67 staining; Fig 3C and 3F) may explain the shortened crypt structures observed by 96 h p.i. of *S*. Tm^WT infection (Fig 3E). Overall, the crypt-villi structure in the cecum of *S*. Tm^WT-infected mice at 96 h p.i. appears disorganized and disrupted in some areas (Fig 3C–F). Of note, at 72 h p.i. *S*. Tm^WT infected wild-type mice showed an intermediate reduction in epithelial proliferation based on Ki67 quantification, falling between the levels observed for *S*. Tm^WT and *S*. Tm^SPI2 at 96 h p.i. (S3C Fig). Conversely, during *S*. Tm^SPI2-infection, increased epithelial cell expulsion at 96 h compared to 24 h occurs alongside enhanced proliferation. Together, this may help to maintain the epithelial barrier structure and results in elongated crypts as observed at 96 h p.i. in the *S*. Tm^SPI2-infected mice (Fig 3C and 3E). Strikingly, TNF levels measured in cecum tissue were significantly elevated in *S*. Tm^WT-infected mice compared to those infected with *S*. Tm^SPI2 (S3D Fig), consistent with earlier studies proposing TNF as a key mediator of epithelial barrier disruption during *S*. Tm^WT infection [23].

To determine whether the effect of TTSS-2 virulence on the epithelium is mediated by the needle complex itself or by translocated effector proteins, we compared 96 h infections with *S*. Tm^SPI2, which lacks a functional TTSS-2 needle, and *S*. Tm^eff, which retains an intact needle complex but lacks almost all known SPI-2 effector proteins [33,34]. Ki67 staining revealed no significant differences in epithelial proliferation between the two strains after 96 h, indicating that the observed TTSS-2–dependent effects are driven by translocated effector proteins (S4A–C Fig).

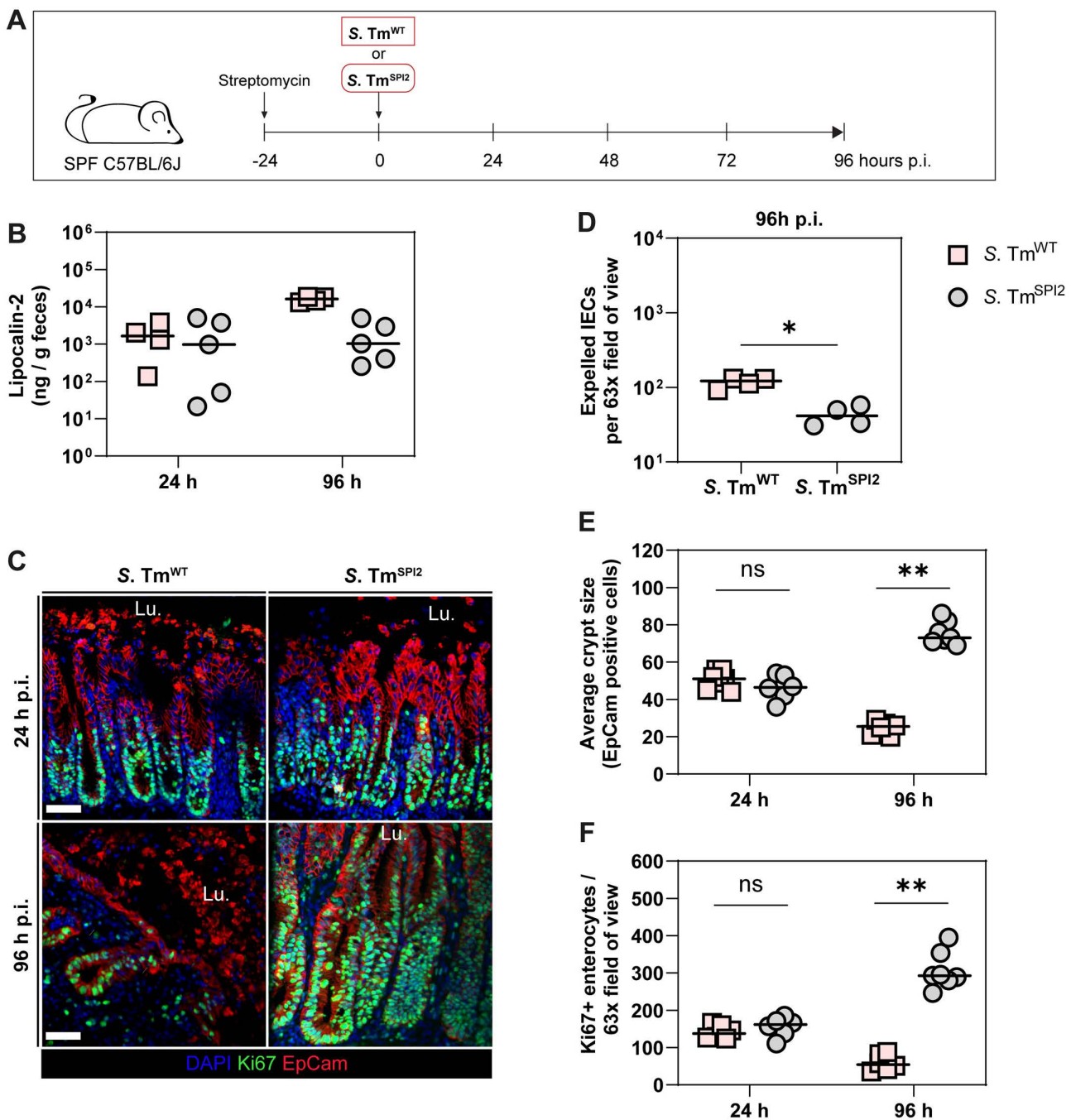

**Fig 3. In wild-type C57BL/6J mice, TTSS-2–dependent inflammation leads to delayed epithelial barrier disruption, occurring by 96 h of *S*. Tm infection.** (a–f) Wildtype C57BL/6J mice were infected with *S*. Tm^WT (SB300) or *S*. Tm^SPI2 (Δ*ssaV*) and sacrificed at 96h p.i. (a) Experimental setup. (b) Luminal Lipocalin-2 levels per gram feces over time as indicator for gut inflammation. (c) Representative micrographs of cecum tissue sections after 24 h and 96 h of *S*. Tm^WT or *S*. Tm^SPI2 infection, stained for cell proliferation marker Ki-67. Lu.: Lumen. Scale bar: 50 µm. (d) Microscopy-based quantification of dislodged enterocytes per 63× field of view in (c). (e) Microscopy-based quantification of average enterocytes per crypt in (c). (f) Microscopy-based quantification of Ki-67 positive epithelial cells per 63x field of view in (c). (b, d-f) Each point represents average of one mouse. Line at median. Mann-Whitney *U* test (*p < 0.05, **p < 0.01, ns – not significant).

In summary, the presence of TTSS-2 virulence exacerbates inflammatory tissue pathology by 96 h p.i., while its absence results in a milder, yet distinct, inflammatory response. These data suggest that mucosal disease severity upon *S.* Tm infection arises from a complex interplay between host immune responses, bacterial burden, and tissue damage.

## Comparison of *S.* Tm^WT- and *S.* Tm^SPI2-infection reveals distinct mucosal gene expression profiles

Our earlier data established a correlation between TTSS-2 virulence and the severity of mucosal histopathology in *S.* Tm-infected mice. However, whether this histopathological difference is reflected at the transcriptional level remained unclear. To investigate if and how TTSS-2 virulence influences broader inflammatory responses beyond epithelial damage, we performed RNA-seq analysis on cecum tissue collected from *S.* Tm^WT- and *S.* Tm^SPI2-infected mice at 24 h and 96 h p.i. As negative controls, we included streptomycin-pretreated, uninfected and *S.* Tm^AVIR-infected mice. The latter mutant fails to invade the mucosal tissue but reaches luminal pathogen loads comparable to *S.* Tm^WT and *S.* Tm^SPI2 [15]. The *S.* Tm^AVIR strain was included, as its inability to invade host tissue prevents the induction of gut inflammation. Therefore, it serves to account for the passive presence of *S.* Tm in the gut lumen, allowing us to distinguish genes regulated by colonization alone from those driven by tissue invasion and active inflammation.

The multidimensional scaling (MDS) plot shows variation among transcription profiles of cecum tissue of the different groups (Fig 4A). All samples passed RNA quality control, with RIN values ≥8 (except for one outlier) (S5A Fig), confirming that observed differences reflect biological rather than technical variation. *S.* Tm^AVIR-infected mice cluster together with streptomycin-pretreated, uninfected mice, indicating that *S.* Tm colonization of the gut lumen without active tissue invasion or pathogen proliferation in the gut tissue does not lead to major differences in gene transcription in the cecum tissue. Interestingly, both 24 h and 96 h *S.* Tm^SPI2-infection cluster together with 24h *S.* Tm^WT-infection, while 96 h *S.* Tm^WT-infected mucosal tissue cluster separately from all other groups (Fig 4A). This transcriptional separation supports our microscopy findings (Fig 3C–F), suggesting that the late-stage response to *S.* Tm^WT infection (96 h p.i.) is qualitatively distinct from early *S.* Tm^WT infection (24 h p.i.) and from both early and late responses to *S.* Tm^SPI2 infection.

Strikingly, numerous genes are differentially expressed between *S.* Tm^WT- and *S.* Tm^SPI2- infections at 96 h p.i. (Fig 4B and 4C). Biological pathway analysis showed that differentially expressed genes (DEGs) are involved in, i.e., cell adhesion, immune responses, chemotaxis, extracellular matrix organization, tissue remodelling and neutrophil chemotaxis (S5B and S5C Fig). qPCR analysis confirmed increased expression of pro-inflammatory genes (TNFα, IL-1β, IL-6, and IFNγ) in *S.* Tm^WT- compared to *S.* Tm^SPI2-infected cecum tissue at 96 h p.i. (S5D Fig). In contrast, the comparison at 24 h p.i. revealed markedly fewer DEGs between *S.* Tm^WT- and *S.* Tm^SPI2- infections (S6A and S6B Fig), with enriched biological processes primarily related to immunoglobulin production (S6C and S6D Fig). These findings underscore the role of TTSS-2 in driving inflammatory responses at later stages of *S.* Tm infection, as transcriptomic differences are substantially less pronounced at 24 h p.i.

To further characterize the immune cell infiltration underlying these transcriptional changes, we performed immunofluorescence staining for Ly6B.2 (a neutrophil marker) and F4/80 (a macrophage marker). At 24 h p.i., there were no significant differences between *S.* Tm^WT and *S.* Tm^SPI2 infections in terms of Ly6B.2+ or F4/80+ cells (S7A–D Fig). However, by 96 h, the lamina propria of *S.* Tm^WT-infected mice exhibited higher numbers of Ly6B.2+ cells than *S.* Tm^SPI2-infected tissues (Fig 4D and 4E). F4/80+ cells (e.g., macrophages) were also more abundant during *S.* Tm^WT infection at 96 h (Fig 4F and 4G). Interestingly, while neutrophils were often seen transmigrating into the gut lumen, macrophages remained within the tissue at both time points (S7E and S7F Fig).

In summary, by 96 h p.i., *S.* Tm induces cecum inflammation regardless of the presence or absence of TTSS-2 virulence. However, TTSS-2 virulence in *S.* Tm^WT-infected mice elicits an amplified immune response, characterized by increased phagocyte recruitment, which is milder in *S.* Tm^SPI2-infected mice and, importantly, transcriptionally distinct. These differences suggest TTSS-2-dependent immune defence regulation may drive disease exacerbation in *S.* Tm^WT-infected mice.

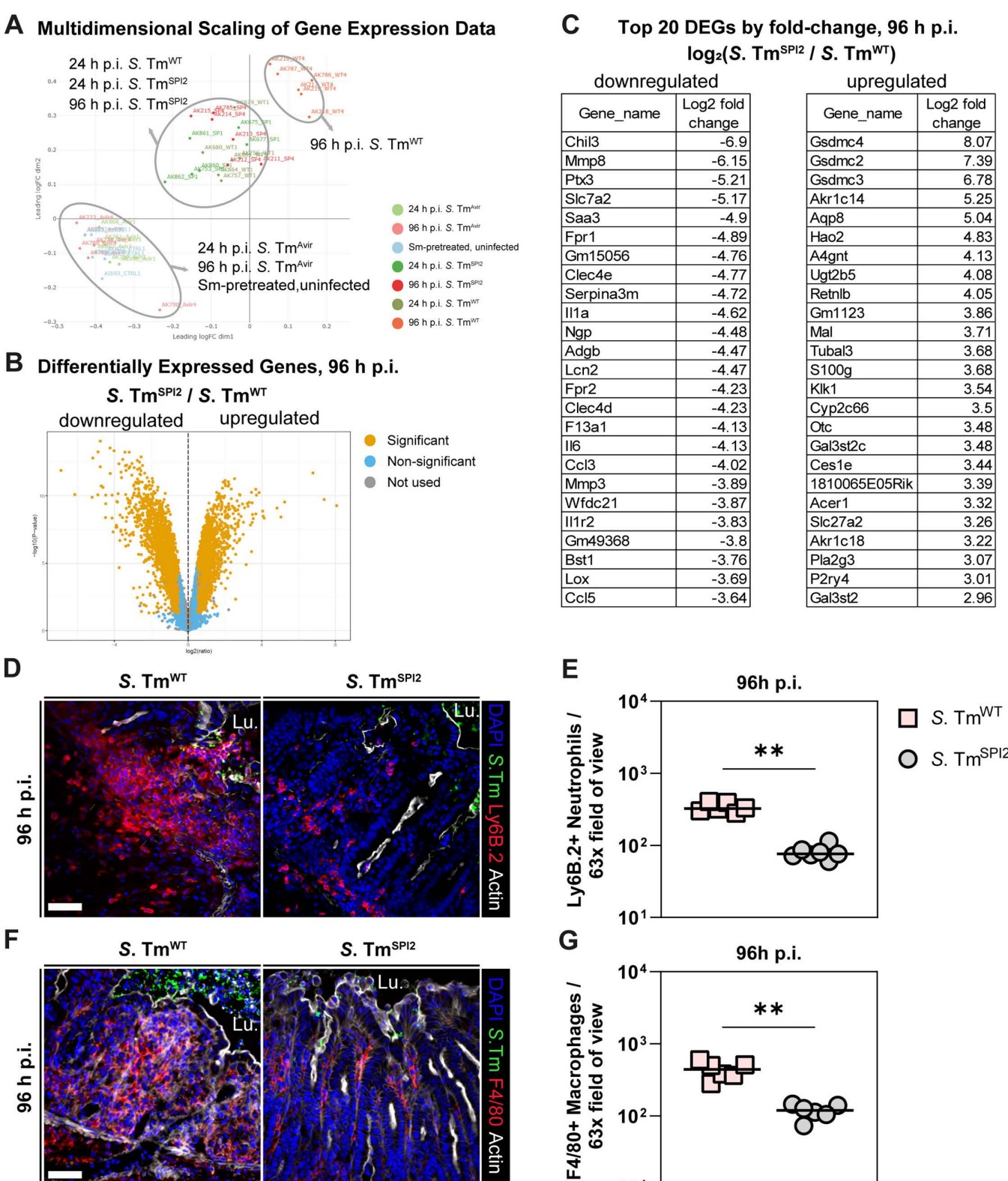

**A  Multidimensional Scaling of Gene Expression Data**

24 h p.i. *S.* Tm^WT
24 h p.i. *S.* Tm^SPI2
96 h p.i. *S.* Tm^SPI2

96 h p.i. *S.* Tm^WT

24 h p.i. *S.* Tm^Avir
96 h p.i. *S.* Tm^Avir
Sm-pretreated,uninfected

Leading logFC dim2

Leading logFC dim1

- 24 h p.i. *S.* Tm^Avir
- 96 h p.i. *S.* Tm^Avir
- Sm-pretreated, uninfected
- 24 h p.i. *S.* Tm^SPI2
- 96 h p.i. *S.* Tm^SPI2
- 24 h p.i. *S.* Tm^WT
- 96 h p.i. *S.* Tm^WT

**B  Differentially Expressed Genes, 96 h p.i.**

*S.* Tm^SPI2 / *S.* Tm^WT

downregulated          upregulated

-log10(P-value)

log2(ratio)

- Significant
- Non-significant
- Not used

**C  Top 20 DEGs by fold-change, 96 h p.i.**

log₂(*S.* Tm^SPI2 / *S.* Tm^WT)

| downregulated | | upregulated | |
|---|---|---|---|
| Gene_name | Log2 fold change | Gene_name | Log2 fold change |
| Chil3 | -6.9 | Gsdmc4 | 8.07 |
| Mmp8 | -6.15 | Gsdmc2 | 7.39 |
| Ptx3 | -5.21 | Gsdmc3 | 6.78 |
| Slc7a2 | -5.17 | Akr1c14 | 5.25 |
| Saa3 | -4.9 | Aqp8 | 5.04 |
| Fpr1 | -4.89 | Hao2 | 4.83 |
| Gm15056 | -4.76 | A4gnt | 4.13 |
| Clec4e | -4.77 | Ugt2b5 | 4.08 |
| Serpina3m | -4.72 | Retnlb | 4.05 |
| Il1a | -4.62 | Gm1123 | 3.86 |
| Ngp | -4.48 | Mal | 3.71 |
| Adgb | -4.47 | Tubal3 | 3.68 |
| Lcn2 | -4.47 | S100g | 3.68 |
| Fpr2 | -4.23 | Klk1 | 3.54 |
| Clec4d | -4.23 | Cyp2c66 | 3.5 |
| F13a1 | -4.13 | Otc | 3.48 |
| Il6 | -4.13 | Gal3st2c | 3.48 |
| Ccl3 | -4.02 | Ces1e | 3.44 |
| Mmp3 | -3.89 | 1810065E05Rik | 3.39 |
| Wfdc21 | -3.87 | Acer1 | 3.32 |
| Il1r2 | -3.83 | Slc27a2 | 3.26 |
| Gm49368 | -3.8 | Akr1c18 | 3.22 |
| Bst1 | -3.76 | Pla2g3 | 3.07 |
| Lox | -3.69 | P2ry4 | 3.01 |
| Ccl5 | -3.64 | Gal3st2 | 2.96 |

**D**

*S.* Tm^WT          *S.* Tm^SPI2

96 h p.i.

Lu.                    Lu.

DAPI *S.*Tm Ly6B.2 Actin

**E**

**96h p.i.**

Ly6B.2+ Neutrophils / 63x field of view

**

- *S.* Tm^WT
- *S.* Tm^SPI2

**F**

*S.* Tm^WT          *S.* Tm^SPI2

96 h p.i.

Lu.                    Lu.

DAPI *S.*Tm F4/80 Actin

**G**

**96h p.i.**

F4/80+ Macrophages / 63x field of view

**

**Fig 4.  *S.* Tm^WT- and *S.* Tm^SPI2-infection elicit distinct mucosal gene expression profiles by 96 h p.i. in C57BL/6J mice.** (a) Multidimensional scaling (MDS) of normalized gene expression data. Each point represents one sample. (b) Volcano plot showing the differentially expressed genes (DEGs)

between *S.* Tm$^{SPI2}$ and *S.* Tm$^{WT}$ at 96 h p.i. The x-axis represents the log$_2$ fold change in gene expression (*S.* Tm$^{SPI2}$ vs. *S.* Tm$^{WT}$), and the y-axis shows the −log$_{10}$ of the p-value. (c) List of the top 20 most strongly upregulated and top 20 most strongly downregulated genes between *S.* Tm$^{SPI2}$ and *S.* Tm$^{WT}$ at 96 h p.i., selected from the 500 most significant DEGs. Genes are ranked by absolute log$_2$ fold change. Positive fold change values indicate higher relative expression in *S.* Tm$^{SPI2}$, and negative values indicate higher relative expression in *S.* Tm$^{WT}$. (d) Representative micrographs of cecum tissue sections after 96h of *S.* Tm$^{WT}$ or *S.* Tm$^{SPI2}$ infection, stained for neutrophil marker Ly6B.2. Lu.: Lumen. Scale bar: 50 μm. (e) Microscopy-based quantification of Ly6B.2 positive cells per 63x field of view in (d). (f) Representative micrographs of cecum tissue sections after 96 h of *S.* Tm$^{WT}$ or *S.* Tm$^{SPI2}$ infection, stained for macrophage marker F4/80. Lu.: Lumen. Scale bar: 50 μm. (g) Microscopy-based quantification of F4/80 positive cells per 63x field of view in (f). (e, g) Each point represents average of one mouse. Line at median. Mann-Whitney *U* test (*p < 0.05, **p < 0.01, ns – not significant).

### *GsdmC* expression is downregulated during TTSS-2 mediated gut inflammation, but GSDMC-deficiency does not alter disease pathology

Among the most differentially expressed genes identified in the RNA-seq analysis were *GsdmC1-4* genes (Figs 4C and 5A). In mice, *GsdmC1-4* are four genes orthologous to the human *GsdmC* gene that encode for Gasdermins C1-4 (GSDMC1–4) [45]. GSDMC belongs to the Gasdermin family, in mice consisting of GSDMA1–3, GSDMC1–4, GSDMD and GSDME, which are pore-forming proteins that drive lytic cell death and inflammation [46]. GSDMD was shown to be the only Gasdermin that affects *S.* Tm pathogen loads in streptomycin-pretreated mice [47]. GSDMD is a downstream effector of NLRC4 and, like NLRC4-deficiency, GSDMD-deficiency in mice leads to the collapse of the epithelial barrier after 72 h of *S.* Tm$^{WT}$ infection [47]. In contrast to GSDMD, GSDMC-deficiency does not have an impact on *S.* Tm loads in streptomycin-pretreated mice (at 48h p.i.) [47] and its role during *S.* Tm infection remains poorly defined. However, a recent study implicated involvement of GSDMC in gut damage and repair [48], prompting us to re-examine whether GSDMC might contribute to epithelial barrier disruption driven by *S.* Tm's TTSS-2 virulence.

While normalized expression levels of *GsdmC1-4* are similar between *S.* Tm$^{WT}$ and *S.* Tm$^{SPI2}$ infection at 24 h p.i., the expression of the genes is strongly downregulated in *S.* Tm$^{WT}$, but not in *S.* Tm$^{SPI2}$ infected mice, at 96 h p.i. based on RNAseq and qPCR analysis (Figs 5A and S8A). Based on this observation we hypothesized that the absence of GSDMC could contribute to the tissue disruptive phenotype we see at 96 h of *S.* Tm$^{WT}$ infection. To investigate this, we used GSDMC-deficient mice that lack the Genes *GsdmC1-4* entirely [47]. We infected GSDMC-deficient mice and their heterozygous littermate controls with *S.* Tm$^{SPI2}$ for 96 h and analysed the state of the cecum tissue by microscopy.

To this end, we pretreated the mice with ampicillin, following the protocol established by Gül et al. [38], which demonstrated that ampicillin pretreatment allows *S.* Tm$^{SPI2}$ to induce strong intestinal inflammation. Notably, gut colonization by *S.* Tm$^{SPI2}$ after ampicillin pretreatment is comparable to that observed with streptomycin pretreatment and to colonization by *S.* Tm$^{WT}$ [38]. While both groups showed similar luminal colonization by *S.* Tm$^{SPI2}$ (S8B Fig), we did observe no difference in disease severity between GSDMC-deficient mice and GSDMC-proficient controls, as assessed by tissue regeneration capacity (Ki67 + cells) at 96 h p.i. (Fig 5B and 5C).

These findings suggest that while reduced *GsdmC1-4* expression correlates with severe, tissue disruptive inflammation, it is insufficient to explain the observed differences in disease progression (between *S.* Tm$^{WT}$ and *S.* Tm$^{SPI2}$). Moreover, its absence does not lead to the collapse of the epithelial layer in *S.* Tm$^{SPI2}$-infected mice. Thus, *GsdmC1-4* expression does not appear to be a critical driver of disease severity in *S.* Tm$^{SPI2}$-infected mice and is insufficient to explain why *S.* Tm$^{SPI2}$ fails to elicit wild-type levels of mucosal damage by 96 h p.i.

### Discussion

It is well established that both TTSS-1 and TTSS-2 virulence contribute to gut inflammation [15,40,49]. Earlier studies showed that mice infected with *S.* Tm$^{SPI2}$ develop moderate and less severe inflammation compared to mice infected with *S.* Tm$^{WT}$, suggesting that SPI-2 virulence contributes to more severe gut inflammation [15,40]. However, it remained unclear how SPI-2 virulence influences the progression of inflammatory pathology, particularly at later stages of *Salmonella* infection. Our data underscore that TTSS-2 virulence correlates with *S.* Tm-induced inflammatory epithelial barrier

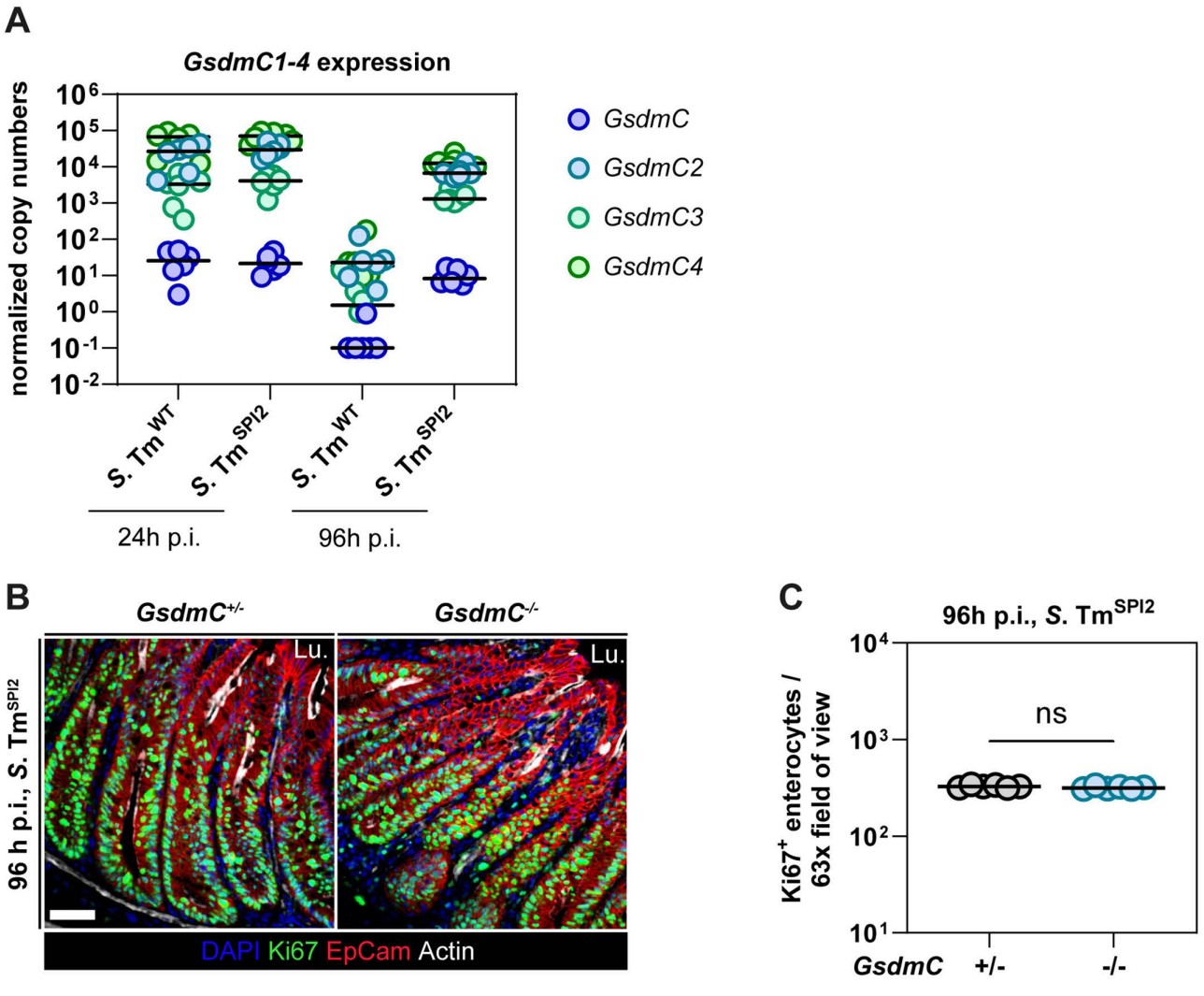

**Fig 5. *GsdmC* expression is downregulated during TTSS-2 mediated gut inflammation, but GSDMC-deficiency does not alter disease pathology at 96 h p.i.** (a) Normalized copy numbers of *GsdmC1-4* RNA of *S.* Tm$^{SPI2}$- and *S.* Tm$^{WT}$-infected C57BL/6 J mice at 24 h and 96 h p.i. (b, c) *GsdmC$^{+/-}$* and *GsdmC$^{-/-}$* mice were Ampicillin-pretreated, infected with *S.* Tm$^{SPI2}$ and sacrificed at 96 h p.i. (b) Representative micrographs of cecum tissue sections, stained for cell proliferation marker Ki-67 at 96 h p.i. Lu.: Lumen. Scale bar: 50 μm. (c) Microscopy-based quantification of Ki-67 positive epithelial cells per 63x field of view in (b). Each point represents average of one mouse. Line at median. Mann-Whitney *U* test (*p < 0.05, **p < 0.01, ns – not significant).

collapse at later stages of infection and that a lack of functional TTSS-2 virulence prevents epithelial barrier collapse in immunodeficient hosts.

To further understand this association, a previous study compared *S.* Tm strains lacking either TTSS-1 or TTSS-2 in wildtype mice [15]. However, since TTSS-1 is essential for efficient mucosal invasion, using TTSS-1 mutants may not accurately reflect the natural infection route – these mutants passively access the lamina propria slower via dendritic cell sampling [16]. Inflammation in such models occurs later and is driven by SPI-2. In contrast, by comparing *S.* Tm$^{WT}$ and *S.* Tm$^{SPI2}$ (both using the normal TTSS-1–dependent invasion), our study enables a more direct assessment of TTSS-2's contribution to inflammation at 72–96 h p.i. Our findings align with previous studies suggesting the involvement of SPI-2

virulence in gut inflammation [15,40,41,49] while offering a more realistic model since the normal invasion route of *S*. Tm is reflected.

It is thought that severe, tissue damaging colitis is a consequence of dysregulated immune defences in response to an overt pathogenic insult [50]. More recent studies started to disentangle how mucosal immune deficiencies contribute to inflammation and gut epithelial barrier disruption. For example, studies employing wild-type *S*. Tm infections showed that the epithelial NAIP/NLRC4 inflammasome and downstream effectors (e.g., GSDMD) are crucial for preventing destructive inflammation in 72 h *S*. Tm$^{WT}$ infection [23,47]. In our present study, we examined destructive colitis from two perspectives: pathogen virulence and host immune competence. Specifically, we asked whether severe gut inflammation in immune-deficient hosts (NLRC4- or CYBB-deficient) could still be triggered by infection with an attenuated strain, *S*. Tm$^{SPI2}$. NLRC4-deficient mice lack a crucial first line defence mechanism in IECs, resulting in accelerated infection kinetics and a rapid accumulation of pathogen-associated molecular patterns (PAMPs) that provoke broad immune responses such as TNF release [20–23]. Notably, epithelial barrier collapse occurred in NLRC4-deficient mice only after *S*. Tm$^{WT}$ infection (as shown in [23]), but not following *S*. Tm$^{SPI2}$ infection at 72 h p.i. (present work). To test whether this observation holds true also in other immune deficient mice, we tested CYBB-deficient mice in the same experimental set up. *Cybb* encodes the NADPH oxidase complex, which is essential for generating antimicrobial reactive oxygen species and plays a key role in controlling infections by *S*. Tm, *Mycobacterium tuberculosis*, or *Aspergillus* spp. Strikingly, the same pattern as in NLRC4-deficient mice was observed in CYBB-deficient mice: 72 h after *S*. Tm$^{WT}$ infection the epithelial barrier collapsed while after 72 h of *S*. Tm$^{SPI2}$ infection the epithelial barrier remained intact. However, for ethical reasons we did not assess enough mice to fully back this up by statistical methods. Interestingly, the absence of the important NADPH oxidase defence did not lead to an increase of pathogenicity of *S*. Tm$^{SPI2}$. One possible explanation could reside in the inability of *S*. Tm$^{SPI2}$ to achieve similar gut-tissue densities as *S*. Tm$^{WT}$ [15,26–29,31–35], since we observed correlation between higher cecum tissue *S*. Tm counts and reduced epithelial regeneration capacity (S2F Fig). In summary, our data build upon previous findings demonstrating that the collapse of the epithelial barrier in immunodeficient hosts is strongly associated with the employment of *S*. Tm's TTSS-2 virulence and its role in promoting pathogen survival or growth in host tissues.

The question remained whether epithelial barrier collapse is a phenomenon restricted to immunodeficient hosts. To address this, we compared the effects of *S*. Tm$^{WT}$ and *S*. Tm$^{SPI2}$ infection in wild-type C57BL/6J mice. Our findings revealed that epithelial barrier collapse also occurs in immunocompetent wild-type C57BL/6J mice, but only following infection with *S*. Tm$^{WT}$ after 96 h. In contrast, *S*. Tm$^{SPI2}$ infection did not trigger epithelial barrier collapse, even at this late stage (96 h p.i.). Notably, at 72 h p.i. with *S*. Tm$^{WT}$, wild-type C57BL/6J mice exhibited an intermediate reduction in epithelial proliferative capacity, suggesting that the pathology develops more gradually in wild-type mice. This delay compared to immunodeficient (NLRC4- or CYBB-deficient) mice indicates that immune competence mitigates early epithelial damage but does not ultimately prevent barrier breakdown. These results indicate that TTSS-2 virulence, rather than immune deficiency (like NLRC4- or CYBB-deficiency) per se, is the critical driver of barrier collapse. Nonetheless, barrier disruption occurred earlier in NLRC4- and CYBB-deficient mice (72 h p.i.), suggesting that these immune defects accelerate disease progression. Together, these data highlight the detrimental effects of TTSS-2 virulence in exacerbating enteropathy in the murine gut. While our study is conducted in a murine model, it is important to note that several aspects of *Salmonella*-intestinal epithelium interactions differ between mice and humans. For example, key components of innate immune sensing – such as NAIP/NLRC4 inflammasome specificity – differ between mice and humans [21,22,51–55]. Moreover, studies using human-derived enteroid systems have shown that epithelial responses to *S*. Tm can differ from those observed in mice, highlighting that not all host–pathogen interactions are conserved across species [53,54,56]. Thus, while our conclusions specifically relate to the inflamed murine cecum, future work in human-derived models will be important to assess how TTSS-2–dependent effects translate to human enteric infection. Emerging organoid systems that integrate immune cells and higher-order tissue architecture may be particularly valuable in this regard [53,57], as they more closely

recapitulate human intestinal physiology and could therefore provide a more complete picture of TTSS-2–mediated host–pathogen interactions.

Epithelial morphology of *S.* Tm$^{WT}$-infected cecum tissue at 96 h p.i. displayed hallmark signs of epithelial barrier collapse, including crypt shortening, increased IEC extrusion and reduced IEC proliferation. In contrast, *S.* Tm$^{SPI2}$-infected tissue exhibited elongated crypts, elevated IEC proliferation and ongoing extrusion, suggesting preserved epithelial integrity despite inflammation. These findings highlight distinct phenotypic outcomes in response to *S.* Tm$^{WT}$ versus *S.* Tm$^{SPI2}$ infection. Despite these morphological differences, both groups exhibited high inflammation according to fecal Lcn-2 levels (above $10^3$ ng/g feces, [39]), with only slightly elevated levels in *S.* Tm$^{WT}$-infected mice. Lcn-2, an acute-phase protein secreted primarily by neutrophils, macrophages, and epithelial cells in response to inflammatory stimuli [58,59], was not sufficient to clearly discriminate between the two infection types. Even though we observed different numbers of Ly6B.2-positive (common neutrophil marker) and F4/80-positive cells (Macrophage/monocyte marker), Lcn-2 levels only slightly differed between *S.* Tm$^{WT}$ and *S.* Tm$^{SPI2}$ infected mice at 96 h p.i. Thus, while Lcn-2 is a robust marker of gut inflammation, it does not sufficiently reflect the qualitative differences in inflammatory responses driven by TTSS-2 virulence. Comprehensive assessment of the inflammatory state likely requires integration of multiple histological and cellular parameters.

For a detailed characterization of TTSS-2-dependent inflammation, we performed transcriptome analysis of whole cecum tissue at 24 h p.i. and at 96 h p.i. comparing *S.* Tm$^{WT}$- and *S.* Tm$^{SPI2}$-infected mice. While gene expression at 24 h p.i. was comparable between *S.* Tm$^{WT}$ and *S.* Tm$^{SPI2}$, about 6000 genes were significantly differentially up or down regulated between the groups at 96 h p.i. The DEGs reflected the distinct epithelial states following infection with *S.* Tm$^{WT}$ and *S.* Tm$^{SPI2}$. Many genes were upregulated more strongly in *S.* Tm$^{WT}$ compared to *S.* Tm$^{SPI2}$ infected mice. Such upregulated genes included genes associated with macrophages and neutrophils or *Mmp3* and *Mmp8* that are involved in tissue remodelling [60]. Strikingly, at 96 h p.i. *GsdmC1-4* mRNA was almost completely depleted in *S.* Tm$^{WT}$ infected cecum tissue, in contrast to *S.* Tm$^{SPI2}$ infection. This suggested a potential role of GSDMC1–4 in modulating the severity of gut inflammation. Notably, the downregulation of *GsdmC1-4* genes has previously been reported in highly inflamed gut tissue in a DSS colitis mouse model [61]. The role of GSDMC in the gut remains poorly understood; however, recent studies have linked it to immune defence against helminth infections. It has been proposed that upregulation of GSDMC in epithelial cells may contribute to both intestinal inflammation and anti-helminth responses [61–64]. Yet, little is known about the role of GSDMC in bacterial infections. In fact, it was shown that GSDMC does not influence pathogen loads during acute *S.* Tm$^{WT}$-induced gut infection (48h p.i.) [47]. We aimed to disentangle a potential involvement of GSDMC in modulating gut inflammation by analysing infection and inflammation kinetics of GSDMC-deficient mice infected for 96 h with *S.* Tm$^{SPI2}$. However, we could not observe any changes in inflammation kinetics and severity (i.e., regeneration capacity) between GSDMC-deficient and -proficient mice in *S.* Tm$^{SPI2}$ infections. Thus, our results suggest that the absence of GSDMC alone does not render the murine cecum susceptible to destructive inflammation and epithelial barrier collapse. However, it is possible that GSDMC downregulation is a secondary effect of *S.* Tm$^{WT}$ infection or that its impact may only become evident at later stages of infection. Alternatively, a combined loss of GSDMC and other factors (i.e., in conjunction with the collapse of the cecum epithelium) might be required to reveal a functional role. This is a question for further research.

In summary, our study highlights the profound impact of TTSS-2 virulence on the nature and severity of gut inflammation at 96 h p.i., revealing striking differences between *S.* Tm$^{WT}$- and *S.* Tm$^{SPI2}$-induced responses. Even though these experiments were performed in streptomycin-pretreated mice, residual microbiota and their metabolites might still influence *S.* Tm virulence and host responses. Therefore, our findings should be interpreted with the caveat that microbial context may contribute to the observed inflammatory differences. We partially mitigate this uncertainty by using littermates that share similar microbiota composition; however, this cannot fully exclude subtle microbiota-driven effects. Importantly, the distinction between *S.* Tm$^{WT}$- and *S.* Tm$^{SPI2}$-induced inflammation is consistently observed across experiments,

suggesting that the major differences reported here are driven by TTSS-2–dependent mechanisms rather than stochastic variation in residual commensals. This distinction between S. Tm^WT- and S. Tm^SPI2-induced inflammation is particularly important given that S. Tm^SPI2, despite having markedly different effects on the host environment than S. Tm^WT, is widely used in *Salmonella* research, e.g., to enable long-term, non-lethal infections in C57BL/6 mice [39,43,65]. For example, recent findings suggest that *Salmonella*'s ability to exploit inflammation via the Tsr chemotaxis receptor is evident in S. Tm^WT-infected mouse guts but not in S. Tm^SPI2 infections [41]. This is likely due to TTSS-2–dependent modulation of the gut luminal niche, as shown by competition experiments where even in the presence of inflammation-inducing S. Tm^SPI2, avirulent Tsr^+ strains outcompete Tsr^- counterparts [41]. This example illustrates the altered inflammatory environment in TTSS-2–deficient infections (rather than proposing a direct TTSS-2–dependent functionality). The distinct inflammatory microenvironments elicited by wild-type S. Tm and TTSS-2 mutants – such as differences in IEC expulsion or neutrophil infiltration – are thus likely to shape a variety of inflammation-dependent phenotypes in *Salmonella*-infected mice. Therefore, the decision to use either S. Tm^WT or S. Tm^SPI2 in experimental setups is not merely technical but may affect the outcome of host-pathogen interaction studies and the interpretation of these data.

## Materials and methods

### Ethics statement

All experiments were conducted in compliance with ethical and legal regulations and received approval from the Cantonal Veterinary Office Zürich under the licenses ZH193/2016, ZH158/2019, ZH108/2022, and ZH109/2022.

### Mouse lines

The experiments were conducted using 8–14-week-old male or female mice. The sample size was not predetermined, and the mice were randomly allocated to different groups. Wild type mice were descendants of C57BL/6 (breeders originally sourced from Jackson laboratories) and have a standard complex microbiota (specific pathogen-free; SPF). All genetic modified mice were of C57BL/6 background. Specifically, the following mouse lines were used: *Nlrc4*^-/- [66], *Cybb*^-/- [67], *GsdmC*^-/- [47]. All mice were bred in controlled conditions in individually ventilated cages at the EPIC and RCHCI mouse facilities of ETH Zurich in Switzerland.

### Bacterial strains and growth conditions

This study used bacterial strains of *Salmonella* Typhimurium SL1344 (S. Tm^WT; SB300; SmR) [68] and mutant derivatives (S. Tm^SPI2; M2730; SmR and M2730; SmR; AmpR) [41,69].

For in vivo infections, S. Tm strains were cultured in lysogeny broth (LB) containing respective antibiotics (50 µg/ml streptomycin (AppliChem)) overnight at 37°C before sub-culturing in 1:20 LB without antibiotics for 4 h. The inoculum was washed and reconstituted in cold PBS (BioConcept).

In some experiments (Figs 3, 4, S3, S5, S6 and S7), two additional avirulent S. Tm (S. Tm^Avir,TSR+ and S. Tm^Avir,TSR-) strains [41] were included in the inoculum at 20-fold lower concentrations than the main infectious strain. This was done to allow concurrent screening for unrelated projects using the same animals. These supplemental strains did not influence tissue pathology and are therefore not considered in the interpretation of the present data. All strains used are summarized in Table 1.

### Infection experiments

Infection experiments in antibiotic-pretreated mice were carried out following the well-established Streptomycin mouse model for oral S. Tm infection, which conventionally focuses on the cecum for analysis, as it represents the main S. Tm invasion site in this model [1]. Briefly, mice received 25 mg of streptomycin via oral gavage 24 hours before infection, and

**Table 1. Bacterial strains used in this study.**

| Strain name used in the study | Strain number | Relevant genotype | Resistance* | Reference |
|---|---|---|---|---|
| *S*. Tm^WT | SB300 | Wild-type, *S*. Tm SL1344 | Sm | Hoiseth et al., 1981 |
| *S*. Tm^SPI2 | M2730 | SL1344 Δ*ssaV* | Sm | Periaswamy et al., 2012 |
| *S*. Tm^eff | NA170 | SL1344 Δ*spvR* Δ*pipAB* Δ*pipB2* Δ*gtgA* Δ*sifB* Δ*gtgEsseI* Δ*steBsseJ* Δ*pipD* Δ*sseL* Δ*gogBsteE* Δ*sopD2* Δ*slrp* Δ*steA* Δ*steDsseK2* Δ*sspH2* Δ*sseK3* Δ*steC* Δ*cigR* Δ*sseK1* Δ*srfJ* Δ*sseFG* Δ*sifA* | Sm | Chen et al., 2021 |
| *S*. Tm^SPI2 | Z6847 | SL1344 Δ*ssaV* | Sm, Amp | Gül et al., 2024 |
| *S*. Tm^Avir | M2702 | SL1344 Δ*ssaV; ΔinvG* | Sm | Periaswamy et al., 2012 |
| *S*. Tm^Avir,TSR+ | Z6832 | SL1344 Δ*ssaV; ΔinvG* | Sm, Km | Gül et al., 2024 |
| *S*. Tm^Avir,TSR- | Z6837 | SL1344 Δ*ssaV; ΔinvG; Δtsr* | Sm, Cm | Gül et al., 2024 |

* Relevant resistances only: Sm = ≥50 µg/ml streptomycin; Cm = ≥15 µg/ml chloramphenicol; Km = ≥50 µg/ml kanamycin; Amp = ≥100 µg/ml ampicillin.

on day 0, they were orally inoculated with $5 \times 10^7$ CFU of *S*. Tm. Fecal samples were collected at specified time points, and cecum tissue and mesenteric lymph nodes were collected at the end of the infection.

For cecum tissue analysis, a gentamycin protection assay was employed to eliminate extracellular bacteria. The cecum tissue was sliced longitudinally, quickly rinsed three times in PBS, and incubated for 45–90 minutes at room temperature in PBS containing 400 µg/ml gentamycin (Sigma-Aldrich). Following incubation, the tissue was thoroughly washed three times (30 sec each) in PBS before plating. Cecum bacterial loads were determined as CFU per total cecum tissue, following standard practice [1].

To plate the samples, tissues were homogenized with a steel ball in a Qiagen tissue lyser – 2.5 min at 25 Hz. The homogenates were serially diluted in PBS and plated on MacConkey agar (Oxoid) supplemented with streptomycin and then incubated overnight at 37°C. Colony-forming units (CFU) were counted the following day and reported as CFU per gram of sample content or per organ.

## Immunofluorescence microscopy

Frozen cecal tissues were sectioned at 10 µm using a microtome (Microm HM525, Thermo Scientific) and mounted on glass slides. Sections were outlined with an ImmEdge pen, rehydrated in PBS (5 min), permeabilized with 0.5% Triton X-100/PBS (10 min), and blocked in 10% NGS/PBS (≥30 min). The antibodies used for staining included α-*S*.Tm LPS (O-antigen group B factor 4–5, Difco) at a 1:400 dilution, α-EpCam/CD326 (clone G8.8, Biolegend) at a 1:400 dilution, α-Ly6B.2 clone 7/4 (BioRad) at 1:200, α-F4/80 (Lucerna Chem AG) at 1:400 and α-Ki67 (Abcam Biochemicals) at 1:400 in combination with suitable secondary antibodies (α-rabbit-Alexa-Fluor488 (Abcam Biochemicals) 1:400; α-rat-Cy3 (Jackson) 1:400) and fluorescent probes, such as AlexaFluor647-conjugated Phalloidin (Molecular Probes) and DAPI (Sigma Aldrich). They were then incubated with primary antibodies in 10% NGS/PBS (≥40 min), washed in PBS (3×), followed by secondary antibody and staining incubation (≥40 min), washed again (3×), and mounted with Mowiol and a coverslip.

Microscopy analysis was done with confocal fluorescence microscopy (Zeiss Axiovert 200M microscope) and with widefield fluorescence microscopy (Axioplan 2 imaging microscope). Images were processed using VisiView (Visitron) and Fiji (ImageJ). Quantitative analysis using widefield microscopy included counting of Ki67+, F4/80+ and Ly6B.2+ cells per 63x field of view. Dislodged IECs were defined as cells located above the epithelial monolayer and detached from the crypt–villus axis, identified based on DAPI and EpCam staining. Representative images illustrating the criteria for dislodged versus non-dislodged IECs are provided in (S9 Fig).

## RNA sequencing

Cecum tissue samples were snap frozen in RNAlater, thawed on ice and transferred to RLT buffer (RNeasy Kit, QIAGEN). Samples were homogenized using 5 mm steel beads in a TissueLyser (QIAGEN) for 3 minutes at 25 Hz. After centrifugation, the supernatant was mixed with 70% ethanol, and RNA extraction was completed according to the manufacturer's protocol (RNeasy Kit, QIAGEN).

Extracted RNA samples were submitted to the Functional Genomics Center Zurich for library preparation, sequencing, and downstream data analysis.

## Lipocalin-2 ELISA

Fecal Lcn-2 levels were measured using a commercial ELISA kit (R&D Systems) following the manufacturer's instructions. Briefly, fecal pellets were suspended in PBS and either used undiluted or diluted 1:20 or 1:400. Concentrations were calculated using a four-parameter logistic (4PL) curve in GraphPad Prism version 8.

## TNF ELISA

Cecal tissue samples were extensively washed in PBS before homogenizing in PBS/0.5%Tergitol (Sigma-Aldrich, Chemie Brunschwig AG) with added protease inhibitor cocktail (Roche). TNF concentration was determined with TNF alpha Mouse Uncoated ELISA Kit (Thermo Fisher Scientific) according to the manufacturer's protocol. TNF concentration was normalized to total protein using Bradford assay (Thermo Fisher Scientific).

## RT-qPCR

Cecum tissue samples were flash frozen in RNAlater (Invitrogen) and stored at -80°C until processing. RNA was extracted using RNeasy Mini Kit (Qiagen) and converted to cDNA using RT2 HT First Strand cDNA Kit (Qiagen). qPCR was performed using FastStart Universal SYBR Green Master reagents (Roche) and Ct values were recorded with QuantStudio 7 Flex FStepOne Plus Cycler. Primers were purchased as validated primer assays from Qiagen or previously published primer sequences [39] were used.

## Statistical analysis

To assess statistical significance, nonparametric statistical testing (two-tailed Mann Whitney-U test) for mouse experiments was performed with GraphPad Prism 8 for Windows. P values of $p \leq 0.05$ not significant (ns), $p < 0.05$ (*), $p < 0.01$ (**) and $p < 0.001$ (***).

## Supporting information

**S1 Fig.** (a–f) Experimental setup as described in Fig 1. (a, b) Fecal *S*. Tm^WT (a) and *S*. Tm^SPI2 (b) pathogen loads as determined using MacConkey plates with selective antibiotics. (c) *S*. Tm^WT and *S*. Tm^SPI2 pathogen loads in cecum tissue and (d) in mesenteric lymph nodes. (g) Representative images of H&E-stained cecum tissue sections Lu. lumen, S.E. submucosa edema. Scale bars: 100 μm. (h) Quantification of pathology scores based on histological evaluation of (g). (a-d, h) Each point represents average of one mouse. Line at median. Mann-Whitney *U* test (*p < 0.05, **p < 0.01, ns – not significant).
(TIF)

**S2 Fig.** (a–f) Experimental setup as described in Fig 2. (a, b) Fecal *S*. Tm^WT (a) and *S*. Tm^SPI2 (b) pathogen loads as determined using MacConkey plates with selective antibiotics. (c) *S*. Tm^WT and *S*. Tm^SPI2 pathogen loads in cecum tissue, (d) in mesenteric lymph nodes and (e) in the spleen. (f) Cecum tissue S. Tm loads plotted over Ki67 + enterocytes/63X

field of view. Plot contains all mice from Figs 1 and 2. Dots filled with red colour belong to mice with reduced regeneration capacity in Figs 1C and 2C. (a-f) Each point represents average of one mouse. Line at median. Mann-Whitney $U$ test (*p < 0.05, **p < 0.01, ns – not significant).
(TIF)

**S3 Fig.** (a–d) Experimental setup as described in Fig 3. (a) Fecal $S$. Tm$^{WT}$ and $S$. Tm$^{SPI2}$ pathogen loads as determined using MacConkey plates with selective antibiotics. (b) $S$. Tm$^{WT}$ and $S$. Tm$^{SPI2}$ pathogen loads in cecum tissue. (c) Dots with grey border are re-plotted from Fig 3E for reference. Microscopy-based quantification of Ki-67 positive epithelial cells per 63x field of view (72 h p.i. and 96 h p.i.) (d) TNFα concentrations were quantified by ELISA and expressed relative to total protein content (pg/mg), as determined by Bradford assay. (a-d) Each point represents average of one mouse. Line at median. Mann-Whitney $U$ test (*p < 0.05, **p < 0.01, ns – not significant).
(TIF)

**S4 Fig.** (a–c) Experimental setup as described in Fig 3. Dots with grey border are re-plotted from S3 Fig for reference. (a) Fecal $S$. Tm$^{WT}$, $S$. Tm$^{SPI2}$ and $S$. Tm$^{eff}$ pathogen loads as determined using MacConkey plates with selective antibiotics. (b) $S$. Tm$^{WT}$, $S$. Tm$^{SPI2}$ and $S$. Tm$^{eff}$ pathogen loads in cecum tissue. (c) Microscopy-based quantification of Ki-67 positive epithelial cells per 63x field of view (a-c) Each point represents average of one mouse. Line at median. Mann-Whitney $U$ test (*p < 0.05, **p < 0.01, ns – not significant).
(TIF)

**S5 Fig.** (a) RIN values of individual RNA samples before library preparation. All samples exceeded the quality threshold (RIN ≥ 8) required for sequencing, except for one sample with RIN = 3.9. (b, c) Dot plot of enriched Gene Ontology (GO) biological processes among differentially expressed genes (DEGs) between $S$. Tm$^{SPI2}$ and $S$. Tm$^{WT}$. The x-axis represents the gene ratio (number of DEGs associated with a GO term divided by the total number of input DEGs). Dot size corresponds to the number of genes in each term, and colour indicates statistical significance (adjusted p-value). (b) 96 h p.i. Only upregulated genes. (c) 96 h p.i. Only downregulated genes. (d) 96 h p.i. Quantification of mRNA expression levels in the cecal mucosa by RT-qPCR. Results are presented relative to b-actin mRNA levels.
(TIF)

**S6 Fig.** (a) Volcano plot showing the DEGs between $S$. Tm$^{SPI2}$ and $S$. Tm$^{WT}$ at 24 h p.i. The x-axis represents the log$_2$ fold change in gene expression ($S$. Tm$^{SPI2}$ vs. $S$. Tm$^{WT}$), and the y-axis shows the –log$_{10}$ of the p-value. (b) List of the top 20 most strongly upregulated and top 20 most strongly downregulated genes between $S$. Tm$^{SPI2}$ and $S$. Tm$^{WT}$ at 24 h p.i., selected from the 500 most significant DEGs. Genes are ranked by absolute log$_2$ fold change. Positive fold change values indicate higher relative expression in $S$. Tm$^{SPI2,}$ and negative values indicate higher relative expression in $S$. Tm$^{WT}$. (c, d) Dot plot of enriched Gene Ontology (GO) biological processes among differentially expressed genes (DEGs) between $S$. Tm$^{SPI2}$ and $S$. Tm$^{WT}$. The x-axis represents the gene ratio (number of DEGs associated with a GO term divided by the total number of input DEGs). Dot size corresponds to the number of genes in each term, and colour indicates statistical significance (adjusted p-value). (c) 24 h p.i. Only upregulated genes. (d) 24 h p.i. Only downregulated genes.
(TIF)

**S7 Fig.** (a) Representative micrographs of cecum tissue sections stained for neutrophil marker Ly6B.2 at 24h p.i. Lu.: Lumen. Scale bar: 50 µm. (b) Microscopy-based quantification of Ly6B.2 positive cells per 63x field of view in (a). (c) Representative micrographs of cecum tissue sections, stained for macrophage marker F4/80 at 24h p.i. Lu.: Lumen. Scale bar: 50 µm. (d) Microscopy-based quantification of F4/80 positive cells per 63x field of view in (c). (e) Representative micrographs of cecum lumen sections, stained for macrophage marker F4/80 at 96h p.i. Lu.: Lumen. Scale bar: 50 µm. (f)

 

Microscopy-based quantification of F4/80 positive cells per 63x field of view in (e). (b, d, f) Each point represents average of one mouse. Line at median. Mann-Whitney $U$ test (*$p<0.05$, **$p<0.01$, ns – not significant).
(TIF)

**S8 Fig.** (a) Quantification of mRNA expression levels in the cecal mucosa by RT-qPCR of $S$. Tm$^{SPI2}$- and $S$. Tm$^{WT}$-infected C57BL/6 J mice at 96 h p.i. Results are presented relative to b-actin mRNA levels. (b) Experimental setup as described in Fig 5. Fecal $S$. Tm$^{SPI2}$ pathogen loads as determined using MacConkey plates with selective antibiotics.
(TIF)

**S9 Fig.** (a) Representative micrographs of cecum lumen sections, stained for EpCam, Actin and DAPI. Ep.: Epithelium. Dashed line: Actin brush border that indicates the outer line of the epithelium. O: Intestinal epithelial cells that are classified as non-dislodged for quantification as they are in contact with the epithelial layer. X: Intestinal epithelial cells that are classified as dislodged as they are located in the lumen without attachment to the epithelial layer. Scale bar: 20 μm.
(TIF)

## Acknowledgments

We would like to thank members of the Hardt lab for helpful discussions. We acknowledge the staff of the ETH Zürich mouse facility EPIC/RCHCI (especially Manuela Graf, Katharina Holzinger, Dennis Mollenhauer, Sven Nowok, Dominik Bacovcin and Emily Müller) and the staff of the Microbiology Institute. The authors gratefully acknowledge the Functional Genomics Centre Zurich (FGCZ) of University of Zurich and ETH Zurich for the support on Genomics analyses. We thank Neal Alto for sharing the SPI-2 effector less $S$. Tm mutant ($S$. Tm$^{eff}$). We thank Manja Barthel and Anna Sintsova for technical assistance. Language editing was done using OpenAI's ChatGPT tool, with all content verified and finalized by the authors.

## Author contributions

**Conceptualization:** Ursina Enz, Ersin Gül, Wolf-Dietrich Hardt.

**Data curation:** Ursina Enz, Ersin Gül.

**Formal analysis:** Ursina Enz, Ersin Gül.

**Funding acquisition:** Wolf-Dietrich Hardt.

**Investigation:** Ursina Enz, Ersin Gül, Luca Maurer, Kamilė Čerepenkaitė, Jemina Huuskonen, Stefan A. Fattinger.

**Methodology:** Ursina Enz, Ersin Gül, Luca Maurer, Kamilė Čerepenkaitė, Jemina Huuskonen, Stefan A. Fattinger.

**Project administration:** Wolf-Dietrich Hardt.

**Supervision:** Ursina Enz, Ersin Gül, Wolf-Dietrich Hardt.

**Validation:** Ursina Enz, Ersin Gül, Luca Maurer, Kamilė Čerepenkaitė, Jemina Huuskonen, Stefan A. Fattinger.

**Visualization:** Ursina Enz, Ersin Gül, Luca Maurer, Kamilė Čerepenkaitė.

**Writing – original draft:** Ursina Enz.

**Writing – review & editing:** Ursina Enz, Ersin Gül, Luca Maurer, Wolf-Dietrich Hardt.

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
