## [Decision Letter · Decision Letter 0]

10 Aug 2025

TTSS-2 virulence drives inflammatory destruction of the gut epithelial barrier and modulates inflammatory response profiles in the *Salmonella* -infected mouse gut

PLOS Pathogens

Dear Dr. Hardt,

Thank you for submitting your manuscript to PLOS Pathogens. After careful consideration, we feel that it has merit but does not fully meet PLOS Pathogens's publication criteria as it currently stands. Therefore, we invite you to submit a revised version of the manuscript that addresses the points raised during the review process.

Please submit your revised manuscript within 60 days Oct 09 2025 11:59PM. If you will need more time than this to complete your revisions, please reply to this message or contact the journal office at plospathogens@plos.org. Please include the following items when submitting your revised manuscript:

We look forward to receiving your revised manuscript.

Kind regards,

Min Yue

Academic Editor

PLOS Pathogens

David Skurnik

Section Editor

PLOS Pathogens

Editor-in-Chief

PLOS Pathogens

orcid.org/0000-0003-2946-9497

Editor-in-Chief

PLOS Pathogens

orcid.org/0000-0002-7699-2064

**Additional Editor Comments:**

The data from a recent resource paper using macrophage infection model may help you to elucidate SPI and associated network cooperation during the infections.

Nat Microbiol 10, 1006–1023 (2025). https://doi.org/10.1038/s41564-025-01953-5

**Journal Requirements:**

3) Some material included in your submission may be copyrighted. According to PLOSu2019s copyright policy, authors who use figures or other material (e.g., graphics, clipart, maps) from another author or copyright holder must demonstrate or obtain permission to publish this material under the Creative Commons Attribution 4.0 International (CC BY 4.0) License used by PLOS journals. Please closely review the details of PLOSu2019s copyright requirements here: PLOS Licenses and Copyright. If you need to request permissions from a copyright holder, you may use PLOS's Copyright Content Permission form.

Potential Copyright Issues:

i) Figures 1A, 2A, and 3A. Please confirm whether you drew the images / clip-art within the figure panels by hand. If you did not draw the images, please provide (a) a link to the source of the images or icons and their license / terms of use; or (b) written permission from the copyright holder to publish the images or icons under our CC BY 4.0 license. Alternatively, you may replace the images with open source alternatives. See these open source resources you may use to replace images / clip-art:

4) Thank you for indicating that "Additional analyses of the data are deposited in Zenodo under embargo and will become available upon publication (https://doi.org/10.5281/zenodo.15783423)." Please note that, though access restrictions are acceptable now, your entire minimal dataset will need to be made freely accessible if your manuscript is accepted for publication. This policy applies to all data except where public deposition would breach compliance with the protocol approved by your research ethics board. 

2) If any authors received a salary from any of your funders, please state which authors and which funders..

6) Thank you for stating "No competing interests." Please modify your Competing Interest statement on the submission form to the standard "The authors have declared that no competing interests exist."

**Reviewers' Comments:**

Reviewer's Responses to Questions

**Part I - Summary**

Reviewer #1: The research by Enz et al. is an elegant study that enhances our understanding of the impact of T3SS-2 on disrupting the epithelial barrier and modulating the inflammatory response in the gut. This is a well-planned and well-written manuscript that closely investigates the role of T3SS-2 in epithelial inflammation and the stability of the epithelial barrier. Through the application of animal models for Salmonella infection, this work provides multiple important insights and, therefore, in my opinion, is worth publishing.

Reviewer #2: The study by Enz et al. provides valuable insights into TTSS-2-mediated gut pathology through rigorous comparison of Salmonella wild-type and TTSS-2 mutant strains across multiple mouse models. The innovative integration of immune-deficient mice model with temporal transcriptomics successfully demonstrates that TTSS-2 is essential for late-stage epithelial barrier collapse and distinct inflammatory microenvironments, which is an advance in understanding virulence-specific host manipulation. However, several conceptual and technical limitations are to be addressed to fully support the conclusions.

**Part II – Major Issues: Key Experiments Required for Acceptance**

Reviewer #1: For the RNASeq experiments, I did not find any information regarding RNA quality assessment (such as gels or RIN). My question is, if there is substantial disruption of the epithelial barrier in the case of STm WT, do the RNAs isolated from this group have comparable quality to those from other groups?

LINES 283-284

“Interestingly, both 24 h and 96 h S. TmSPI2-infection cluster together with 24h S. TmWT-infection, while 96 h S. TmWT-infected mucosal tissue cluster separately from all other groups (Fig 4A).”

Reviewer #2: Major points:

1. The manuscript identifies TTSS-2 as a driver of epithelial barrier disruption and amplified inflammation, but the underlying molecular mechanisms remain unclear. While transcriptomic data reveal differential gene expression (e.g., GsdmC downregulation), the specific TTSS-2 effectors or host signaling pathways (e.g., TNF-mediated responses) linking TTSS-2 activity to epithelial damage are not defined.

2. The study uses NLRC4- and CYBB-deficient mice to assess immune deficiency impacts, but this experimental design may need to be better explained. It is unclear whether other immune defects would yield similar TTSS-2 dependency, leaving the broader relevance of the observed phenotypes uncertain.

3. While testing GSDMC-KO with S. TmSPI2 was a logical first step based on the RNA-seq data, the more critical experiment would be to infect GSDMC-KO mice with S. TmWT to see if the absence of GSDMC attenuates the severe barrier disruption phenotype observed at 96h. Besides, could GSDMC downregulation work synergistically with other pathways altered in WT infection (e.g., TNF signaling)? The manuscript mentions this possibility but doesn't explore it experimentally.

4. Although the "threshold density" concept (Lines 143–144, Fig S2I) links bacterial load to epithelial destruction, the manuscript does not explore how TTSS-2 enables surpassing this threshold. Intracellular replication assays may mechanistically anchor this observation.

5. The timing of TTSS-2-mediated effects (72 h in immune-deficient vs. 96 h in wild-type mice) is noted, but the basis for this delay is not explained. Besides, the temporal analysis of barrier disruption provides valuable new evidence of delayed pathology in wild-type mice, but the 96hpi focus misses earlier TTSS-2 effector activity windows. Kinetic assessments (<72h) would strengthen the claim that structural collapse follows progressive barrier dysfunction.

**Part III – Minor Issues: Editorial and Data Presentation Modifications**

Reviewer #1: The authors compare (S Fig. 1, S Fig. 2, etc.) different mouse models but not directly between the STm WT and STmSPI2. Throughout the text, they compare the mutant strain to the WT. In my opinion, it is worth rearranging the supplementary figures to compare two variables simultaneously—mouse strain and Salmonella strain—and applying multiple comparison statistical tests. This would allow readers to easily see the full differences in pathogen behavior and improve clarity. There is data on CFU/cecum tissue, but since there is no data on cecum weight, which may vary significantly between animals, it would be beneficial to report CFU per gram of tissue.

Given the significantly different organ loads (MLN, spleen) when comparing mouse strains, it would also be useful to compare Salmonella variants. Since the work focuses on the role of T3SS-2 in intestinal barrier stability and inflammation, clearly showing differences between STm and STmSPI2 would be advantageous.

Reviewer #2: Minor points:

1. While the chemotaxis exploitation hypothesis presents an intriguing dimension of TTSS-2 functionality, direct evidence linking reduced neutrophil recruitment in SPI-2 infections (Fig 4E) to impaired luminal colonization is absent.

2. The assessment of gut inflammation relies heavily on Lcn-2 levels and Ki67 staining, but additional markers would strengthen phenotypic characterization.

3. Besides cecum, other intestinal regions (e.g., colon, ileum) are not assessed, leaving uncertainty about whether TTSS-2 effects are cecum-specific.

4. The sample sizes for CYBB experiments (Fig 2C, E; n=3-4) may lead to unreliable conclusions, necessitating either statistical justification for the current sample size or an increase in the number of specimens.

5. Cytokine analysis is limited to TNFα (Fig S3C); measuring additional cytokines such as IL-1β, IFNγ in cecal lysates would provide a more comprehensive picture of the inflammatory response.

6. RNA-seq identified DEGs (such as GsdmC in Fig 4C) would require more validation via qPCR or WB due to their mechanistic significance.

7. Unclear rationale for including S. TmAvir strains in transcriptomics (Lines 275-277) without results discussion.

8. When quantifying "dislodged enterocytes" (Fig 3D), the criteria for defining an "expelled IEC" (e.g., morphological features or marker co-staining) are not specified, making the quantification method ambiguous.

PLOS authors have the option to publish the peer review history of their article (what does this mean? ). If published, this will include your full peer review and any attached files.

**Do you want your identity to be public for this peer review?** For information about this choice, including consent withdrawal, please see our Privacy Policy .

Reviewer #1: No

Reviewer #2: No

**Figure resubmission:**

**Reproducibility:**



---

## [Editor Report · Decision Letter 1]

14 Nov 2025

Dear Prof. Hardt,

We are pleased to inform you that your manuscript 'TTSS-2 virulence drives inflammatory destruction of the gut epithelial barrier and modulates inflammatory response profiles in the *Salmonella* -infected mouse gut' has been provisionally accepted for publication in PLOS Pathogens.

Additionally, I would request a few minor changes to further enhance discussion section of this study, see below.

Best regards,

Min Yue

Academic Editor

PLOS Pathogens

David Skurnik

Section Editor

PLOS Pathogens

Sumita Bhaduri-McIntosh

Editor-in-Chief

PLOS Pathogens

orcid.org/0000-0003-2946-9497

Michael Malim

Editor-in-Chief

PLOS Pathogens

orcid.org/0000-0002-7699-2064

Editor Comments:

The study investigates how Salmonella enterica serovar Typhimurium (S. Tm) uses its Type III Secretion System 2 (TTSS-2, encoded by SPI-2) to manipulate host responses in the mouse gut during orogastric infection. Using antibiotic-pretreated C57BL/6J mice (wild-type and immune-deficient models like NLRC4-/- and CYBB-/-), the authors compare wild-type S. Tm (S. TmWT) with a TTSS-2 mutant (S. TmSPI2, ΔssaV).

The work advances understanding of TTSS-2 as a modulator of gut pathology, beyond its known role in intracellular survival. It aligns with prior studies showing TTSS-2 effectors (e.g., SteD, SpvB) suppress host immunity and promote replication in macrophages.

While reviewers focused on mechanisms and design, several relevant aspects were not addressed, which may serve as new ways to further enhance the quality of the manuscript.

1. Antibiotic pretreatment depletes microbiota, but TTSS-2 may interact with residual commensals (e.g., via competition or dysbiosis amplification). Recent studies show microbiota modulates S. Tm virulence (e.g., Bacteroides influencing TTSS-2 expression). Without 16S rRNA profiling or germ-free controls, claims on "inflammatory microenvironments" may overlook microbial confounders. Please add discussion with relevant literature on this point, considering microbiome context in the gut is another key factor for the observed pathology.

2. Mouse models are standard, but there is no mention of human enteropathy (e.g., nontyphoidal salmonellosis). TTSS-2 orthologs in human pathogens vary, and mouse-specific inflammasomes (e.g., NLRC4) differ from human NAIP/NLRC4. A 2021 paper (mBio. 2021 Jan 12;12(1):e02684-20) on S. Tm in human enteroids showed TTSS-2-independent barrier effects at low doses.
---

## [Editor Report · Acceptance letter]

Dear Prof. Hardt,

We are delighted to inform you that your manuscript, "TTSS-2 virulence drives inflammatory destruction of the gut epithelial barrier and modulates inflammatory response profiles in the *Salmonella* -infected mouse gut," has been formally accepted for publication in PLOS Pathogens.

Best regards,

Sumita Bhaduri-McIntosh

Editor-in-Chief

PLOS Pathogens

orcid.org/0000-0003-2946-9497

Michael Malim

Editor-in-Chief

PLOS Pathogens

orcid.org/0000-0002-7699-2064